# A New PCR-Based Assay for Testing Bronchoalveolar Lavage Fluid Samples from Patients with Suspected *Pneumocystis jirovecii* Pneumonia

**DOI:** 10.3390/jof7090681

**Published:** 2021-08-24

**Authors:** Flora Marzia Liotti, Brunella Posteraro, Giulia De Angelis, Riccardo Torelli, Elena De Carolis, Domenico Speziale, Giulia Menchinelli, Teresa Spanu, Maurizio Sanguinetti

**Affiliations:** 1Dipartimento di Scienze Biotecnologiche di Base, Cliniche Intensivologiche e Perioperatorie, Università Cattolica del Sacro Cuore, Largo A. Gemelli 8, 00168 Roma, Italy; floramarzialiotti@gmail.com (F.M.L.); brunella.posteraro@unicatt.it (B.P.); giulia.deangelis78@gmail.com (G.D.A.); giulia.menchinelli@unicatt.it (G.M.); teresa.spanu@unicatt.it (T.S.); 2Dipartimento di Scienze di Laboratorio e Infettivologiche, Fondazione Policlinico Universitario A. Gemelli IRCCS, Largo A. Gemelli 8, 00168 Roma, Italy; riccardo.torelli@policlinicogemelli.it (R.T.); elena.decarolis@policlinicogemelli.it (E.D.C.); domenico.speziale@policlinicogemelli.it (D.S.); 3Dipartimento di Scienze Mediche e Chirurgiche, Fondazione Policlinico Universitario A. Gemelli IRCCS, Largo A. Gemelli 8, 00168 Roma, Italy

**Keywords:** BAL fluid, dihydrofolate reductase gene, *Pneumocystis jirovecii* pneumonia, qPCR assay

## Abstract

To support the clinical laboratory diagnosis of *Pneumocystis jirovecii* (*PJ*) pneumonia (PCP), an invasive fungal infection mainly occurring in HIV-negative patients, in-house or commercial *PJ*-specific real-time quantitative PCR (qPCR) assays are todays’ reliable options. The performance of these assays depends on the type of *PJ* gene (multi-copy mitochondrial versus single-copy nuclear) targeted by the assay. We described the development of a *PJ*-PCR assay targeting the dihydrofolate reductase (DHFR)-encoding gene. After delineating its analytical performance, the *PJ*-PCR assay was used to test bronchoalveolar lavage (BAL) fluid samples from 200 patients (only seven were HIV positive) with suspected PCP. Of 211 BAL fluid samples, 18 (8.5%) were positive and 193 (91.5%) were negative by *PJ*-PCR. Of 18 *PJ*-PCR-positive samples, 11 (61.1%) tested positive and seven (38.9%) tested negative with the immunofluorescence assay (IFA). All (100%) of the 193 *PJ*-PCR-negative samples were IFA negative. Based on IFA/PCR results, patients were, respectively, classified as having (*n* = 18) and not having (*n* = 182) proven (*PJ*-PCR+/IFA+) or probable (*PJ*-PCR+/IFA−) PCP. For 182 patients without PCP, alternative infectious or non-infectious etiologies were identified. Our *PJ*-PCR assay was at least equivalent to IFA, fostering studies aimed at defining a qPCR-based standard for PCP diagnosis in the future.

## 1. Introduction

Fungal infections, mainly *Pneumocystis jirovecii* (*PJ*; pneumocystosis), *Aspergillus* (aspergillosis), or *Mucorales* (mucormycosis), account for about 5% of all severe respiratory infections (including bacterial, mycobacterial, or viral infections) that occur in immunocompromised patients [1]. The non-specificity of either respiratory symptoms or imaging features necessitates differentiation among possible infectious etiologies as well as between infectious and non-infectious respiratory abnormalities [1]. Molecular-based methods for detecting pneumonia agents in multiple or single assays have been developed to overcome the limits of conventional (microscopy- or culture-based) identification methods, which are familiar in medical mycology [2] especially with *PJ* organisms [3].

While FDA-cleared molecular panels comprise a variety of detectable bacterial or viral agents from the lower respiratory tract (LRT) bronchoalveolar lavage (BAL) fluid—the preferred sample for PCP diagnosis [4]—or other samples [5], only one panel—to our knowledge—has recently comprised *PJ* [6]. Thus, it is unsurprising that in-house or commercial *PJ*-specific PCR assays, mostly real-time quantitative PCR (qPCR) assays, became reliable options to support today’s clinical laboratory diagnosis of *Pneumocystis* pneumonia (PCP) [7]. Despite helping to distinguish colonization from infection, qPCR assays show variable performance [8], which depends on the type of *PJ* gene—multi-copy mitochondrial (i.e., mitochondrial small subunit (mtSSU) of ribosomal RNA) versus single-copy nuclear (e.g., beta-tubulin or dihydrofolate reductase (DHFR)) encoding gene—targeted by the assay. However, the predictably best analytical sensitivity of the mtSSU gene may be detrimental when considering the risk for false-positive results. Notably, quantification of a single-copy gene may better reflect the *PJ* load—in terms of genomic copies/mL—in an LRT sample [8].

Here, we report on the detection of *PJ* DHFR DNA in BAL fluid samples, prospectively collected from patients hospitalized at our institution (a large tertiary care hospital in Rome, Italy), using a new in-house qPCR assay (hereafter referred to as *PJ*-PCR assay).

## 2. Materials and Methods

Between 1 March 2019 and 1 March 2020, patients hospitalized at the Fondazione Policlinico Universitario A. Gemelli IRCCS (Roma, Italy) with a sampled BAL fluid for microbiological examination were eligible for the study. We included samples from patients (age, ≥18 years) with PCP- [9] or non-PCP [1]-compatible symptoms (e.g., cough, dyspnea, and hypoxemia) and radiological findings (e.g., bilateral ground-glass opacities) who had or did not have typical risk factors for PCP [10,11]. Before examination, BAL fluid samples were centrifuged and the resuspended pellets (200 µL) were used for both *PJ*-microscopy (50 µL) and PCR assay (100 µL), as specified below. All samples allowed one to extract enough human DNA to make *PJ* detectable in BAL fluid, based on the positive amplification of the human RNase gene (see below for details). This gene was equivalent to the human β-globin gene used elsewhere.

The immunofluorescence assay (IFA) of BAL fluid samples was performed using the MONOFLUO™ *Pneumocystis jirovecii* IFA Test kit (BioRad, Hercules, CA) to identify *PJ* asci and trophic forms at microscopy. Using the forward (5′-GGCTGATCAAAGAAGCATGGATA) or reverse (5′-CGGCATAGACATATTCGATACTTGTT) primer pair and the internal probe (5′-TGCGTGAAACAGATACATGGAGCTCTACCC) targeting the *PJ* DHFR-encoding gene (GenBank Accession number, DQ269976.1), we developed the *PJ*-PCR assay following Minimum Information for the publication of real-time Quantitative PCR Experiments (MIQE) guidelines [12].

To determine the *PJ*-PCR assay’s performance, the full-length *DHFR* gene was amplified, the resulting PCR product was cloned into a pCR 2.1 cloning vector (Zero Blunt^®^ PCR Cloning Kit, Invitrogen, Carlsbad, CA), and the recombinant plasmid DNA concentration was measured with the Qubit™ 4 fluorometer (Thermo Fisher Scientific, Waltham, MA). Using a 1:10 dilution series of *DHFR*-carrying plasmid (ranging from approximately 10^−1^ to 10^9^ copies/mL), a standard (calibration) curve was generated to calculate the amplification efficiency, which was established to be around 99%. Specifically, using the equation PCR efficiency = 10^−1/slope^ − 1, we plotted the logarithm of the initial DNA concentration on the *x*-axis and the quantification cycle (Cq)—also termed threshold cycle (Ct)—on the *y*-axis. Then, we determined the *PJ*-PCR assay’s analytical sensitivity, expressed as the limit of detection (LoD), which was found to be equivalent to 10 copies/mL. The Ct variation (as assessed at 10X LoD) was 0.6, whereas the *PJ*-PCR assay had 100% analytical specificity, as shown elsewhere [13]. Additionally, we verified that the *PJ*-PCR assay was stable by testing positive (DHFR-carrying plasmid) samples in qPCRs after the reagents were subjected to several freezing/thawing cycles.

As previously described [14,15], a diagnostic platform automatically extracted and dispensed microbial DNA into qPCR reagents’ prefilled microwell strips before manually loading them onto the BioRad CFX Thermal Cycler (BioRad, Hercules, CA, USA) qPCR instrument. Primers’ and probe’s concentrations were 400 nM or 150 nM, respectively, whereas thermal cycling conditions consisted of an initial denaturation step at 95 °C for 8 min, followed by 50 cycles of 95 °C for 15 s and 60 °C for 1 min. For each sample, the number of amplification cycles required to produce a positive signal for the *DHFR* gene was expressed as a Ct value; thus, samples with Ct values ≤40 or >40 were scored as *PJ*-PCR positive or negative, respectively. In each run, the aforementioned human RNase gene was included as an external intrinsic control to monitor PCR inhibition, and no-template (sterile water) controls were included to monitor PCR contamination.

In parallel, patients’ serum samples (collected the same day or within ± one day of BAL fluid sampling) were tested with the Fungitell^®^ assay (Associates of Cape Cod, East Falmouth, MA, USA) for 1,3-β-D-glucan (BDG) measurement using a positivity cutoff of 80 pg/mL [11].

According to positive IFA or *PJ*-PCR results, the revised/updated EORTC/MSGERC 2008 consensus definitions of invasive fungal disease [11] allowed one to classify pneumonia episodes as proven (both IFA and *PJ*-PCR positive) or probable (only *PJ*-PCR positive) PCP episodes, respectively. In view of conflicting opinions about the role of BDG in the PCP diagnosis [4,16,17], we only used BDG results to corroborate the results obtained with the *PJ*-PCR assay. Furthermore, microbiological standard-of-care testing results allowed one to classify non-PCP episodes as bacterial, viral, or (outside PCP) fungal pneumonia episodes, respectively.

## 3. Results

As shown in Figure 1, 200 patients with their BAL fluid samples (*n* = 211) were investigated for PCP. Based on both *PJ*-PCR and IFA results, patients were classified as having (*n* = 18) or not having (*n* = 182) proven (PCR+/IFA+) or probable (PCR+/IFA−) PCP [11]. Samples that tested positive with both PCR and IFA had Ct values lower than in samples that tested positive with PCR and negative with IFA.

Table 1 summarizes *PJ*-PCR/IFA results for 211 pneumonia episodes, 191 of which were single and 20 were multiple. One patient had four episodes and eight patients had each two episodes. Of 211 BAL fluid samples studied in total, 18 (8.5%) tested positive (Ct values, 20.6–37.9) and 193 (91.5%) tested negative (Ct values, >40) with the *PJ*-PCR assay.

Among 18 *PJ*-PCR-positive samples, 11 (61.1%) samples (Ct values, 20.6–36.6) were IFA positive and seven (38.9%) samples (Ct values, 30.2–37.9) were IFA negative. A total of 193 (100%) of 193 *PJ*-PCR negative samples were IFA negative. Seventeen (94.4%) of eighteen patients with *PJ*-PCR-positive BAL fluid samples had BDG-positive (315 to >500 pg/mL) serum samples, whereas one patient with a *PJ*-PCR-positive (Ct value, 37.9) but IFA-negative BAL fluid sample had a BDG-negative (<80 pg/mL) serum sample. The last patient was a hematological patient for whom a BAL fluid culture yielded positive results for both *Pseudomonas aeruginosa* and *Stenotrophomonas maltophilia*.

In 41 (21.2%) of 193 *PJ*-PCR/IFA-negative episodes (Figure 1), all patients (36 of whom were immunocompromised) had BDG-positive (82 to >500 pg/mL) serum samples, and their BAL fluid samples were culture positive for *Candida* (16 samples), *Aspergillus* (two samples), and/or bacterial species (15 samples). Seven of eighteen patients with PCP were co-infected with bacteria (e.g., *Legionella pneumophila*, *Mycobacterium tuberculosis*, etc.) and two other patients were colonized with *Aspergillus* or *Candida* (Table 1). Among 182 patients without PCP, infectious (bacterial, viral, *Candida*, or *Aspergillus*) or non-infectious etiologies were identified (Table 1).

## 4. Discussion

The PCP diagnosis remains particularly challenging in HIV-negative immunocompromised or other seriously ill patients, who accounted for 96.5% (193/200) of patients in this study. Only 7 (3.5%) of 200 patients were HIV positive. Excluding 28 episodes from patients with lower suspicion of disease (Table 1), our *PJ*-PCR assay allowed one to diagnose PCP in 18 (9.8%) of 183 episodes from patients who, otherwise, were at high risk of developing disease. In addition to meeting clinical and radiological criteria, these patients had at least one of the EORTC/MSGERC designed host factors [11]. Besides being highly specific [13], our assay appeared to be highly sensitive for detecting *PJ* DNA in BAL fluid samples. All (100%) of the 193 episodes with a negative PCR result were from patients who had a microbiological (i.e., non-*PJ* respiratory infection) or non-microbiological (e.g., non-infectious respiratory abnormality) reason on which the suspicion of PCP was initially based.

The detection of *PJ* microscopically in the BAL fluid (or pulmonary tissue) using conventional (Gomori methenamine silver) or immunofluorescence (specific fluorescent antibody) staining is the current diagnostic criterion for proven PCP [11]. We found that none of samples with a positive IFA result was negative by the *PJ*-PCR assay, whereas seven samples with a positive PCR result were negative by the IFA. These findings would be consistent with the lower Ct values observed in these samples, which might reflect a smaller burden of *PJ* organisms, thus resulting in false-negative results at the microscopic examination. Six of seven patients were patients not living with HIV (Table 1), which is a condition notoriously associated with a larger burden of *PJ* organisms in respiratory samples [3]. Therefore, we did not interpret the seven IFA negative results as falsely PCR positive results but as the results of patients who had benefited from the molecular rather than the microscopic diagnosis of PCP [4].

It is important to recall that, regardless of the type of PCR format employed for testing, a positive or negative PCR result in the BAL fluid should be interpreted according to strict diagnostic criteria [4]. These include clinical, radiological, and laboratory abnormalities that, unfortunately, may suggest different pneumonia etiologies as underlined above. Therefore, a PCR assay—especially a qPCR assay—is the right adjunct to a multifaceted workup to assist the diagnosis of PCP or other invasive fungal infections [1]. Consistently, 2020 EORTC/MSGERC invasive fungal disease definitions consider a PCR-positive respiratory sample (or a BDG-positive serum sample) as the only mycological evidence for probable PCP [11]. In agreement with the PCR assay, 17 patients with PCR-positive BAL fluid samples had serum samples that tested positive for BDG. In disagreement with the PCR assay, serum samples were BDG-positive in 41 patients with PCR-negative BAL fluid samples. Thus, our findings support the EORTC/MSGERC recommendations that require ≥2 consecutive serum samples to be BDG positive to diagnose probable PCP if other etiologies have been ruled out [11]. Conversely, a negative BDG result—that may be potentially false negative because of a low *PJ* load and low BDG release [16]—strongly suggested the absence of PCP in our patients, which was supported by either the BAL fluid sample PCR negativity or the determination of alternative etiologies (Table 1).

Implementing a *PJ* qPCR assay in the clinical microbiology laboratory may have particular value in the context of the ongoing pandemic due to the severe acute respiratory syndrome coronavirus 2 (SARS-CoV-2) that causes coronavirus disease 2019 (COVID-19), because *PJ*—similar to other fungi (*Aspergillus*, *Candida*, or *Mucorales*)—has emerged as a notable coinfection agent in COVID-19 patients [18]. At the time of writing, the *PJ*-PCR assay was integrated into our clinical laboratory’s diagnostic respiratory workflow, which includes multiplex PCR-based panels newly commercialized to detect bacterial and viral agents of syndromic infections [5]. As shown in this study (and before its integration in the laboratory), the assay underwent validation to clearly delineate its technical performance, including the LoD and the amplification yield or control [12].

With regard to *PJ* qPCR assays, the choice of target genes for amplification remains an open issue [7]. Recently, Huh et al. [19] showed the similarity of two commercial assays—one of which was capable of amplifying the mitochondrial large subunit (mtLSU) of a ribosomal RNA gene—in detecting low *PJ* loads in BAL fluids. Very recently, Dellière et al. [20] found that an automated commercial reverse-transcriptase qPCR (~2.2 h workflow) was more sensitive than their in-house qPCR assays (~5 h workflow), which amplified both mtSSU and mtLSU for detecting *PJ* in respiratory samples. Therefore, the use of DHFR in our study might be questionable because of the (expected) lower sensitivity of a single-copy nuclear gene compared to multi-copy mitochondrial genes [8]. However, our study is reminiscent of a strategy reported by Huggett et al. [21] in 2008. Unlike us, the authors used the *PJ* heat shock protein 70 (HSP70) gene to detect *PJ* DNA in BAL fluid samples of HIV-positive patients (132 in total; only 7 were HIV positive in our study). The analytical sensitivity of their HSP70 real-time assay was ~five copies/reaction, but a cutoff value of ~10 copies/reaction was used to show (based on receiver–operator curve analysis) a clinical sensitivity of 98% and specificity of 96% for the diagnosis of PCP [21]. Furthermore, the authors [21] optimized the DNA-extraction PCR step that, if suboptimal, makes a negative PCR result difficult to interpret. We carefully assessed the sample-processing step (i.e., extraction) so that the *PJ*-PCR assay developed by us could represent the natural expansion of an existing molecular diagnostic platform in our laboratory [14], which is now capable of amplifying unique, species-specific genes using DNA automatically extracted from BAL fluid (or other clinical) samples (~4 h workflow). Comparing our in-house assay with commercially available or, specifically, mtSSU- or mtLSU-based assays for *PJ* detection [7,8] was beyond the scope of the present study. In essence, we sought to develop a qPCR assay that was at least equivalent to the IFA and that, ultimately, allowed us to replace microscopic with molecular testing for PCP diagnosis.

In conclusion, our *PJ*-PCR assay proved to be a sensitive and specific diagnostic tool for *PJ* infection in BAL fluid samples, thus emphasizing the increasing role of *PJ* qPCR assays as a clinical laboratory strategy to diagnose PCP in HIV and non-HIV patients [3,7]. We hope that our assay enters large-scale evaluation studies aimed at defining the performance of a standard qPCR-based method for the diagnosis of PCP.

## Figures and Tables

**Figure 1 jof-07-00681-f001:**
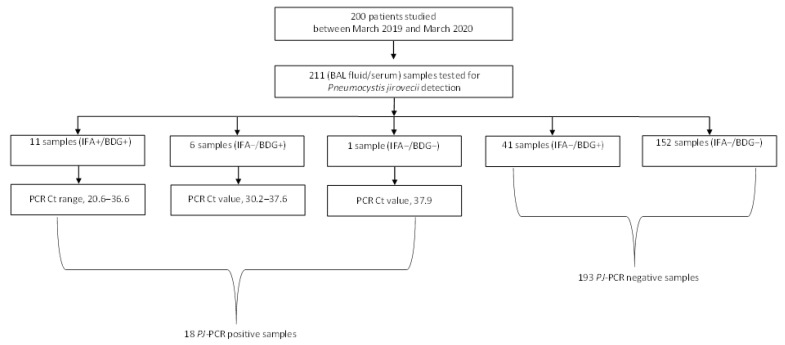
Overview of samples obtained from 200 patients with suspicion of PCP that underwent clinical laboratory testing. Bronchoalveolar lavage (BAL) fluid samples were tested by both *PJ*-PCR and IFA, whereas serum samples, which were obtained almost concomitantly to BAL fluid sampling, were tested for BDG. For *PJ*-PCR positive samples, PCR Ct values are shown according to the positivity (11 samples) or negativity (7 samples) by IFA.

**Table 1 jof-07-00681-t001:** Microbiological results for 211 BAL fluid samples from patients with suspected PCP.

Patient Characteristics	PCR+ (Ct, 30.6 ± 4.4) ^a^ & IFA+ (*n* = 11)	PCR+ (Ct, 35.7 ± 2.6) ^a^ & IFA− (*n* = 7)	PCR− & IFA− (*n* = 193)
Underlying diseases/conditions			
Hematological malignancy	3	1	63
Inflammatory/rheumatic disease	3	4	51
SOT/HSCT ^b^	0	1	31
Solid malignancy	2	0	17
HIV infection	3	1	3
Other disease/condition ^c^	0	0	28
Chest X-ray/CT findings			
Ground-glass opacity (GGO)	5	2	10
Nodules	0	3	7
Consolidation	3	2	112
Other findings ^d^	3	0	64
Infection/colonization due to organisms other than *Pneumocystis jirovecii* ^e^			
Bacteria	4 ^f^	3 ^g^	67
Viruses	0	0	43
*Aspergillus*	0	1 ^h^	7
*Candida*	0	1 ^i^	27
None	7	3	63 ^j^

^a^ Results by *Pneumocystis jirovecii* (*PJ*)-PCR on bronchoalveolar lavage (BAL) fluid samples are expressed as the mean threshold cycle (Ct) value ± standard deviation. Conversely, immunofluorescence assay (IFA) results on the same samples are expressed as positive or negative only. ^b^ SOT, solid organ transplantation; HSCT, hematopoietic stem cell transplantation. ^c^ Includes patients for whom the diagnostic workup comprised *PJ* testing in spite of their apparent lack of underlying conditions/factors known to affect or enhance the exposure to PCP [10]. Twenty-three of these patients were in the intensive care unit (ICU) from the hospital. ^d^ Includes interstitial infiltrates, atelectasis, pleural effusion, or pneumothorax, as assessed by chest X-ray or computed tomography (CT). ^e^ Assessed by semiquantitative culture-, antigen-, or molecular-based standard-of-care testing methods. Some patients were concomitantly infected/colonized by more than one of the listed organisms. With regard to *Candida* organisms, 26 of 28 patients with culture-positive samples were deemed to be colonized by *Candida* species, whereas two remaining patients had a concomitant bloodstream infection caused by *Candida albicans* and, thus, were deemed to be infected by *Candida* species. ^f^ Includes *Haemophilus influenzae* (1 sample), *Legionella pneumophila* (1 sample), methicillin-resistant *Staphylococcus aureus* (1 sample), and methicillin-susceptible *S. aureus* (1 sample). ^g^ Include *Escherichia coli* (1 sample), *Mycobacterium tuberculosis* (1 sample), and *Pseudomonas aeruginosa* plus *Stenotrophomonas maltophilia* (1 sample). ^h^ The sample was positive for *Aspergillus fumigatus*. ^I^ The sample was positive for *Candida albicans*. ^j^ In these episodes, non-infectious etiologies of pneumonia included lung infiltration from the underlying disease, drug-related lung toxicity, etc.

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
