# Peer review of "A New PCR-Based Assay for Testing Bronchoalveolar Lavage Fluid Samples from Patients with Suspected Pneumocystis jirovecii Pneumonia"

_jof, 2021, doi:10.3390/jof7090681_

Round 1

Reviewer 1 Report

The paper by Liotti is well written and adds to the literature but the paper could be improved.

  1. The methods need a sample on BAL processing and what sample was set for IFA vs PCR. Were there some samples where yield was so poor that not even host HPRT could be detected?
  2. It was bit surprising that the authors did not compare their DHFR assay to mtLSU or SSU to compare performance.

Author Response

The paper by Liotti is well written and adds to the literature but the paper could be improved.

  1. The methods need a sample on BAL processing and what sample was set for IFA vs PCR. Were there some samples where yield was so poor that not even host HPRT could be detected?

Answer: We added the required information about the sample processing and about how we controlled the DNA yield in BAL fluid samples to rule out any potential PCR inhibition event. See page 2, lines 68 to 73, and page 3, lines 105 to 107, of the revised manuscript.

  1. It was bit surprising that the authors did not compare their DHFR assay to mtLSU or SSU to compare performance.

Answer: We agree with the reviewer that it may be surprising to not having compared our assay’s performance with that of mtSSU or mtLSU based assays for PJ detection. However, as specified in the manuscript, this issue was beyond the scope of our study. In substance, our goal in this study was to develop a PJ-specific assay that allowed expanding an already existing molecular diagnostic platform in our laboratory. See page 6, lines 236 to 242, of the revised manuscript.

Reviewer 2 Report

The authors are presenting a new qPCR-based assay for Pneumocystis jirovecii pneumonia from the testing BAL fluid samples. The manuscript is already in a good written form and provide important clinical information. The paper is acceptable after major revisions.

-       This paper is very similar to previous published paper (Huggett et al., 2009, Development and evaluation of a real-time PCR assay for detection of Pneumocystis jirovecii DNA in bronchoalveolar lavage fluid of HIV-infected patients) which used HSP70 as target gene. I can’t find significant differences between two papers. What is the major differences?

-       I would like the authors to provide more information about the stability and PCR efficiency of target gene. 

-       Furthermore, to improve the quality of the paper, I would like to see representative phenotypes of P. jirovecii pneumonia in BAL fluid samples.

Author Response

The authors are presenting a new qPCR-based assay for Pneumocystis jirovecii pneumonia from the testing BAL fluid samples. The manuscript is already in a good written form and provide important clinical information. The paper is acceptable after major revisions.

  1. This paper is very similar to previous published paper (Huggett et al., 2009, Development and evaluation of a real-time PCR assay for detection of Pneumocystis jirovecii DNA in bronchoalveolar lavage fluid of HIV-infected patients) which used HSP70 as target gene. I cannot find significant differences between two papers. What is the major differences?

Answer: As required, we added a comment about that our study was reminiscent of a published study by Huggett et al. in 2008 (the relative reference was added), thereby underscoring the major differences between the studies. See page 6, lines 228 to 236, and page 7, lines 316 to 318, of the revised manuscript.

  1. I would like the authors to provide more information about the stability and PCR efficiency of target gene.

Answer: As required, we added relevant information about the stability and efficiency of our PCR assay. See page 2, lines 86 to 91 and 94 to 96, of the revised manuscript.

  1. Furthermore, to improve the quality of the paper, I would like to see representative phenotypes of P. jirovecii pneumonia in BAL fluid samples.

Answer: Unfortunately, no data about PCP phenotypes in BAL fluid samples (e.g., inflammatory markers, neutrophil levels, etc.) that could improve the quality of the paper were available.

Round 2

Reviewer 1 Report

The authors have addressed my concerns. 

Reviewer 2 Report

Now the manuscript is acceptable in this Journal. But I feel the lack of representative phenotype data.